# The Thinking Skills Deficit: What Role Does a Poetry Group Have in Developing Critical Thinking Skills for Adult Lifelong Learners in a Further Education Art College?

**Frances Norton [1,*] and Margaret Gregson [2,*]**

[1] Department of Postgraduate Studies, Leeds Arts University, Leeds LS2 9AQ, UK

[2] Department of Education, Faculty of Education and Society, University of Sunderland, Sunderland SR6 0DD, UK

[*] Correspondence: frances.norton@leeeds-art.ac.uk (F.N.); maggie.gregson@sunderland.ac.uk (M.G.)

**Abstract:** This article investigates Brown's assertion that students today exhibit an unwillingness/inability to engage in critical thinking (CT). He describes this as a 'critical thinking deficit'. The question of whether CT can be taught or whether we can only create the conditions in which CT can thrive and develop is explored through analysis of data from a pedagogical intervention of a Poetry Group; it aims to develop CT by employing Community of Inquiry as a methodology. This intervention was offered to a group of Further Education (FE) students over a period of six months with the intention of preparing them for progression into Higher Education (HE). Findings from the study lend support to the claim that sharing stories and poems is helpful in developing social and cultural capital across the group and in supporting CT and academic development. Students in the study report that they found the Poetry Group particularly valuable in encouraging both critical engagement with their Arts subject, deeper levels of learning and supporting improvements in attainment.

**Keywords:** critical thinking; practice-focused research; Arts Education; Poetry Group; practitioner research; adult lifelong learners

---

## 1. Introduction to the Research

### 1.1. Critical Thinking in Further Education

Participant Lamia states,

> *I think it has encouraged me to think a bit more about what I say and how I say it in here and outside.*

This research asks whether thinking skills can be taught or whether we can only create the conditions in which thinking skills can develop and thrive? It seeks to understand how educators might develop CT skills in the context of a small specialist Arts Education institution. The study aims to explore whether/how a Poetry Group and other related thinking skills pedagogic interventions can develop CT, support progression, encourage skill transfer and improve problem solving competencies for students of the Arts pursuing FE courses.

This research was originally supported by the Learning and Skills Improvement Service (LSIS) in 2012 and more recently by the Education and Training Foundation (ETF) in 2018–2019. The small-

scale research project subsequently led to this larger-scale study. From my professional experiences, I found vocational Further Education students in Arts subjects to be resistant to the written element of the course work, while being enthusiastic about practical sessions. This research seeks to understand whether strategies might be developed to support student progression in written assignments. A broad working hypothesis here is that CT strategies which begin with thoughts expressed in the spoken word, an idea supported by Berger [1], may in turn motivate and encourage students to think and write more critically.

This article introduces the site of the research, conducted at an Arts College (anonymised using British Educational Research Association (BERA) ethical guidelines [2]. Gibb [3] states that students leave education allegedly unable to think for themselves, with few transferable skills, and underprepared for democratic citizenship. Instead, Gibb argues, students should be encouraged to construct knowledge for themselves using CT. Brown [4] (p.2) writes, 'generalizable thinking skills fail the test of transferability between subject matters.' Gibb [3] agrees that subject knowledge is domain specific in the way expertise and insight is taught.

### 1.2. Global Policy on the Critical Thinking Deficit

The Canadian Ministry of Education, Shaheen [5] states that all students will need to develop a flexibility and versatility undreamed of by previous generations. The Foresight Review into the Lifelong Learning [6] (p.5) states,

> *Better skills enable freedom of opportunity, provide people with the tools to adapt to a changing world and promote social mobility, inclusion and wellbeing.*

The Department of Education (DfE) [7] outline issues adult lifelong learners face in returning to education. United Nations Educational Scientific and Cultural Organization (UNESCO) [8] is a policy promoting education and lifelong learning globally. Kennedy, in her Widening Participation publication [9], emphasises the need for adult lifelong learning to reach into the community, and education to work for social cohesion as well as encouraging social mobility.

### 1.3. A Practice-Focused Approach

Practice-focused approaches to educational improvement present an alternative to traditional, passive modes of learning. Barrett and Bolt [10] warn against the normalisation of the passive classroom. This is echoed by Brown [4], who notes that thinking about thinking provides an impetus for remedying, self-defeating rote learning. Sennett [11] states that the path to mastery has stages, beginning with observation, moving to imitation and emulation, then repetition of 10,000 hours to become proficient, and ultimately transcend existing practices to create new ones. This argument is furthered by Oakeshotte [12] who believes that a long period of initiation gives students the time and space to learn and that speaking before students have anything significant to say is learning without understanding. Actions such as debating, writing, speaking and listening develop strategies for allowing what is in a student's mind to get out, Geertz [13]. Practice-focused development allows CT to develop with an initial scaffolding from the facilitator. Participant Matuta describes her initial feelings about the Poetry Group,

> *I think you have to give people the understanding of what critical thinking actually is in the first place and what the benefits of it are. I think once people understand what it is, what it can do for them, then I think they can teach themselves.*

In this way CT concepts of practice based teaching are shown to be formulated through a Community of Inquiry. This will be discussed in the Methodology section of this article 2.4. The Arts Council [14] (p.3) states that a world-class art and design education will engage, inspire and challenge young people, equipping them with the knowledge and skills to participate in, experiment with,

invent and create their own works of art, craft and design. This kind of experimental exploring within education could empower students to think creatively and critically.

*1.4. Is the Critical Thinking Deficit Culturally Specific?*

As explained above, Brown [4] (p.1) contends that students today exhibit a CT deficit. In an article in *Psychology Today*, Camarata [15] (paragraph 2) states that students are adept at memorizing and regurgitating facts. However, reasoning, criticality, and problem solving has been reduced. Camarata puts this down to a misguided attempt to make children 'geniuses' in the 1990s with baby flash cards. Belkin in Camarata [15] believes that students begin with a CT deficit and many graduate without the ability to spot a logical fallacy or construct a cohesive argument. These and other authors argue that "rote learning" approaches to teaching inhibit students' ability to use CT. They claim that "rote learning" and teaching to the test programme students in a way that actually discourages reasoning, CT and problem solving. Camarata [15] (paragraph 6) states that such approaches to education "wire" students from an early age to memorize and retrieve "facts" on demand, but not to think or reason. A lack of CT skills can affect students' personal and professional lives and many university students struggle with real-world problem solving. The concept that CT is very Euro-centric is misplaced. For example, thinkers in China may approach CT traditions via the collective rather than the individualism of Europe, Australia and America, yet not take issue with CT per se. This article raises questions about the effects of institutional, educational, professional and cultural factors on the disposition to think critically. Vandermensbrugghe [16] (p. 421) states that academics need to be more 'interculturally competent'. CT is often problematic for international students. In addition, home students may have the advantage of understanding the language and culture of educational contexts more easily. Vandermensbrugghe [16] (p. 419) goes on to comment that lecturers would benefit from examining learning practices around CT for what they are, socially constructed practice and not superior intellectual practices or a preferred form of cognition that is only accessible to the best (and often most privileged). She notes that in many Asian countries, the group is more important than the individual.

*1.5. Economic Issues Around the Critical Thinking Deficit*

The Arts Council [14] (p.30) states that the UK's creative industries are now worth £84.1 billion per year to the UK economy. In addition, the creative industries employ 1.8 million people and the sector is growing faster than any other industry sector [14] (p.3). They go on to assert that one in eleven of all UK jobs now fall within the creative economy and so critical and creative thinking skills are needed by FE students. UNESCO [17] (paragraph 2), in partnership with the Global Alliance for Literacy, states that literacy is a driver for sustainable development in that it enables greater participation in the labour market, improves child and family health and nutrition, reduces poverty, and expands life opportunities. Increasing CT skills could develop cultural capital and boost employment potential. Baker [18] states that Bourdieu's [19] (pp.241–58) forms of capital describe student narratives which identify enablements and constraints on their social mobility and HE decision-making choices. Participants Vili and Ve are brothers and both are pub chefs, they want to be filmmakers and have come to art college to develop the cultural capital needed to succeed in that industry. Participant Boreas talks of leaving his post-fishing hometown to attend college,

> *I had a need to get it out, like the story was burning in me to be made.*

Many adult lifelong learners at an Arts College opt for the more employable art pathways such as printed textiles or graphic design. CT promotes the development of skill transfer, and social capital and can aid progression to HE or employment in the creative industries.

*1.6. Issues in Further Education*

Broadhead et al. [20] (p. 39), in Perspectives on Access, state that FE in the 1960s and 1970s was a form of radicalisation, challenging the elitism in some universities. Broadhead goes on to state that FE is situated in communities; it is student centred, flexible, and collaboratively practitioner led. Lecturers teaching in FE are passionate about enabling disadvantaged students. Broadhead et al. contend that adult lifelong learners in FE have had to prove themselves worthy of a place in HE. Johnson [21] states that 75% of mental health problems are established by the age of 25. The Office of National Statistics, [22] states that 19% of the UK student population displays common mental health disorders. The potential effects of common mental health disorders are, in the worst cases, suicide and self-harm, as well as use of alcohol and recreational drugs and risk-taking behaviours. Extremism is also an issue for vulnerable students, leading to, for example Far-Right beliefs, violence, and illegal actions in the name of a group. Social, cultural, health, economic, or political factors mean some students do not get a fair opportunity to achieve when at school age, Broadhead et al. [20] (p. 2) the DfE [7] (p.77) concur stating

> *For every learner, there exists a complex and unique relationship between their own perceptions of the personal benefits and personal costs of learning.*

Another issue could be learning differences; dyslexia is very common at Arts Colleges in particular. A study by the Royal College of Art [23] states that 29% of art students identified as dyslexic. Part of the impetus for undertaking this research was my personal difficulties and experience as a dyslexic student of the Arts in the 1980s, and the barriers and frustrations to my own academic learning have been echoed by participants in this study. Participant Roma describes her experience,

> *I am dyslexic. It didn't really affect my writing but it really affected my reading. I really avoided reading. I found reading really frustrating. I felt like 'I'm an idiot' I hate that feeling, I thought I was a moron.*

*1.7. Introduction to the Literature: the Further Education Experience*

There are relatively few papers and books written about CT from the perspective of the FE sector and fewer from FE practitioners—examples include Broad [24]; Broadhead and Gregson [25]; Burke [26]. In a DfE report, Owen [27] (paragraph 6) states that there is a lack of evidence on how current practices operate to improve quality and improve learners' outcomes. This research hopes to highlight this statement in order to address the issue of invisibility of good practice, and academically rigorous practitioner research established in Further Education. The Practitioner Research Programme (PRP) uses the researcher's teaching practice in FE institutions as the inspiration and starting point for practice-based investigation. Practitioner research in the sector in the field of CT may have much to offer in illuminating the key issues in this field of study.

*1.8. A Gap in Knowledge about Critical Thinking in the Arts College*

CT in Arts Education and the concept of thinking through making is especially pertinent for vocational students who work with their hands. There is not a vast amount of research on CT in Further Education or in Arts institutions. The following texts have been helpful in describing the Art School context and in turn helping to articulate the concept of thinking through making. Sennett [11] discusses the desire to make objects well, with integrity, for its own sake and as a template for living. Somerson and Hermano [28] describe students as being immersed in a culture where making questions ideas and objects, using and inventing materials, and activating experience all serve to define a form of CT albeit with one's hands that is "critical making." Lipman [29] judges that CT must be practiced or experienced in tacit, experiential learning. This echoes Aristotelian concept of phronesis, discussed by Broadhead and Gregson [25].

### 1.9. Can Critical Thinking be Taught?

Sennett [11] states that learning a skill is time consuming and takes effort and sacrifice. Elder and Paul [30] have written books, papers and led conferences on the subject. Elder and Paul [30] (p.8), in their *Intellectual Traits* framework, state that critical thinkers are able to question information and points of view. Lipman [29] is adamant that CT also needs creative thinking. Both Elder and Paul and Lipman have a formulaic and commercialised way of teaching CT that demand a kind of absolute adherence to their particular statutes. This calls into question the ontology of their stance. Brown [4] (p.50) states that CT can be taught using 'disciplinary languages'. However, McPeck [31] argues that there are no general thinking skills, since thinking is always thinking about some subject- matter. He qualifies this by writing that CT should not be taught as a separate subject, but that lecturers would be better placed leading students into discussion, argument and cognitive thinking for themselves. This very much matches Lipman's theories about autonomous practice-focused learning, enabling students discover for themselves a path to CT.

### 1.10. Critical Thinking through Practice-Focused Development: Creating the Conditions

Lipman [29] foregrounds the value of practice-focused learning, learning and thinking through doing and situated, embodied practice. hooks [32] discusses practical wisdom as a form of agency and transgression in her book on teaching CT. Holding the Poetry Group in non-formal environments out of teaching hours the group is able to use peer mentoring and creative learning that is not grade chasing but includes personal development and educational enrichment in an informal setting. In this space hooks suggests there is a flattened hierarchy where participants are agential, transgressing the educational norm, participating in a democratic forum where emotion, opinion and thinking is nurtured in a Community of Inquiry. Sennett [11] has serious points to make about practice- based endeavours such as writing and performing poetry. This again links to Aristotle and the Victorian Utilitarianist Mill, making for oneself, a useful, good life, both individually and in community. Dewey [33] continues the theme, considering that the practitioner must care deeply for the subject matter upon which their skill is exercised. As we can see from the above, enduring issues regarding the nature of CT and whether/how it can be taught are a long way from resolution. They do, however, raise some key points in the discourse worthy of further exploration through empirical research. This small-scale research study aims, in its own way, to contribute to this debate.

## 2. Materials and Methods

### 2.1. Methodology, Ontology and Epistemology of this Small Scale Project

Scott and Usher [34] (p.1) advocate practice-led research and point out that this is underpinned by reflecting on wider epistemological and methodological contexts. The ontological underpinning of this study is heuristic, exploring the subjective experience of developing CT with and within a group of students. Context and values are not easily separated. It could be said that both the participants and the researcher are steered by their social locators which shape and construct their thoughts and realities. The epistemology framing the study is interpretivist and the hermeneutic interpretive paradigm is essentially illuminative. Usher [34] (pp.18–22) states that all human interaction is meaningful and is given meaning by interpretive frameworks. The research is grounded in practitioner research (Bell [35] (pp. 8–9)) and informed by the Community of Inquiry theory developed by Biesta [36] and Lipman [29]. In view of the above, this qualitative interpretive study does not employ variables or control groups. The CT intervention used for this study is the Poetry Group. Methods employed in the study include video interviews and field notes.

### 2.2. The Participants

There are 18 participants in the research population—14 women and four men—all from an Arts College, and all from an FE Arts course. They have an age range between 16 and 60 years old. Participants in the research population are from a mix of social and professional backgrounds, with

different levels of educational attainment. All participants are self-selecting volunteers. The data from the Poetry Group forms a small part of a larger PhD study which, to date, has run over two years. In total, there have been five CT interventions and 283 participants. Some of the participants have volunteered in more than one intervention. For example, participant Matuta also takes part in the Debate Club and the Critical Thinking Club, because of the value, group bonding and development she has experienced.

### 2.3. Anthropology

Maynard and Cahnmann-Taylor [37] (p.4) are anthropologists and poets, who use poetry as epistemology, methodology and data analysis. They state that experimental anthropologists prefer more fragmented, post-modern, ethnographic encounters amid "cultural borderlands". These are the place of the creative spirit, not locked into a silo system that education can sometimes encourage, where classification of subject area is closely guarded by those who have invested themselves in the system. The borderlands are an interdisciplinary space, where creative students are free to collaborate and make wider connections. The borderlands exist between education and anthropology; art and education; between practice and research; and between thinking and doing. The borders are the most interesting places holding the tension between educational policy and the creative critical reality of teaching. Maynard and Cahnmann-Taylor [37] state that researchers working in more innovative modes are pushing the bounds of traditional methodologies. Working in an innovative mode is key for arts students.

### 2.4. Community of Inquiry

A Community of Inquiry grows from a group interested in solving a problem together, states McPhail-Bell [38]. In a Community of Inquiry, practice is developed in and through mutual engagement as advocated by Lipman [29] and Biesta [39] who agree that CT involves a conceptual investigation into problematic situations which disturb routine thinking and trigger a process of transaction where thinking occurs across the action in an attempt to understand the nature of the problem and potential responses to it. In this way, problems begin to be unravelled through mutual engagement in a collaborative research process. When this happens in a group, it can become a Community of Inquiry.

### 2.5. Research Design and Approach

Adult lifelong learners enjoy practical aspects of making art. They appear to become disengaged when asked to annotate practice in their sketchbooks, make notes and write an essay. FE is about preparing students for progression either into the creative industries or HE; in either instance, being able to write coherently is useful and necessary. Looking back at 1.6. *Issues in Further Education*, resistance to writing could stem from mental health, ill health, lack of cultural capital or learning differences. My assumption was that art students prefer not to write. In this article, I have devised a CT intervention, the Poetry Group, in order to address these issues.

Denscombe [40] (pp. 3–5) explains that social arts-based researchers are faced with a variety of alternative methods and there are many strategic decisions about which to choose. There is no one right decision, which makes the work of a beginning researcher more complex. In designing this CT intervention, I learned from the other data collection methods used previously in my PhD. I considered a diary project and book club I used three years ago, but discounted both of these because, from experience, the diary only developed writing skills and the book club developed reading and speaking skills. However, a poetry writing Community of Inquiry would, I hoped, develop writing, listening, discussing, reading and speaking skills.

There are three variable elements to the Poetry Group design: use of a published poem, poetry theme and poetry form. The themes for the weekly poetry task were decided by participants as a group and included topics such as art practice, or the five senses illustrated in a poem by Imtiaz Dharker called Tissue [41]. Another theme on dialect led us to poet Liz Berry's Birmingham Roller

[42]. I wrote themes into a timetable assigning each week a theme, a poetic form and an example from published poetry. A handout accompanied the reading of the published poem with the poem printed and a guide as to how to use the poetic form; this was made reference to informally during a discussion of the poem. Young [43] (p.51) states that poetry literature circles help students engage with texts, build verbal communication skills, tap into the power of expression, make personal connections to text, build analytical thinking skills and develop a Community of Inquiry.

Published poems and poetic form were decided by myself; this is warranted by my having written and published poetry for six years, winning a bursary for the Kendal International Poetry Festival; frequenting seminars and poetry writing workshops. I openly led the group not as an expert but an amateur enthusiast—this fit in with the informal learning situation. Emphasis was on informality; poetic form, and use of language and theme was discussed, but using CT was left to the student in terms of how closely they adhered to style and form. Simple poetic forms were chosen such as rhyming couplet, blank verse, tercet and repeated refrain. Below is an excerpt from an interview with student participants Janus and Matuta on the subject of poetic form,

> *FN: Do you think it's helpful having poetry forms to hang a theme off when writing poetry?*
> *Matuta: Yes I like that, it has more interest. It's more of a challenge.*
> *Janus: It focuses your thoughts, instead of having lots of random weird individual thoughts.*

As the Poetry Group is non-formal, some students 'do their own thing', in that they will choose their own theme and poetic form to write a poem. As this was still fulfilling the brief of evolving independent thinking skills, being creative and developing writing skills, this was also a good outcome. The group write a weekly poem and perform it to the group responding to the published poem. After each reading, I facilitated a short peer critique, where the group discussed the participant's poem in a supportive and up-building discussion. Poetry reading and group critique are optional. Writing and performing their poems, participants think about a wider vocabulary, discovering topical and ethical and humorous themes in literature and art. The research design encourages confidence in writing, listening, peer-mentoring, and speaking aloud in front of others. The design was implemented in the hope that a practice of weekly writing for pleasure will spill over into participant's course work. Participants plan in the Community of Inquiry, to collate and produce a self-published zine/pamphlet of a selection of the group's poems.

Divergent experimental thinkers, poets and academics, Retallack and Spahr [44] state that to create a lively, investigative poetry classroom, lecturers must work with contemporary cultural implications of poetry in society. They continue, students must use CT to make meaning of the poetics of our contemporary world, one that is de-centred and pluralistic; multi-cultural, multi-lingual, multi-racial, and multi-contextual. The Poetry Group goes some way to engaging with some of these complex Lyotardian [45] global/local topics, examining the petit narrative of our arts classroom in the context of global issues through poetry.

The research approach is inductive. Campbell-Galman [46] (p.22) explains that 'bottom–up' inductive data sorting uses the data itself to make ideas and theories. Data is collected from participants in a number of ways: from the poems themselves, from video interviews and from field notes. Powell [47] (p. 103) states that video recording is a permanent record; Pirie [48] believes video can be revisited during data analysis. From this data, patterns and themes are interpreted using an 'I' poem data analysis. Kara [49] (p.117–118) states that picking first-person statements out from an interview and arranging them in a list creates a poem by which to spot themes and patterns quickly. Inductive analysis is used to process the data and begins with the particularity of Arts College participants, looking for recurring themes in the 'I 'poem to make inference to more general cases. In the Poetry Group, linguistics and language are a potent truth-telling media. Durrant [50] (p.6.) states that it is empowering for participants to be seen as a whole person, freed from the power dynamics and roles of student and teacher. Within the group, participants speak and write about all aspects of their lives using autoethnography written about by Ellis et al. [51]; also by accessing well-being

through lifeworld care, meaning considering the educative and personal aspects of students' lives (Hemmingway [52]).

*2.6. Ethical Considerations*

BERA guidelines, University of Sunderland and Leeds Arts University ethics approval was given for this research project. The study is representational and avoids deceptions in dealings with participants who have an information sheet and sign a participant agreement form. They are free to leave the study at any point. The research aims to protect and anonymise participants and institutions.

## 3. Results

*3.1. Data Interpretation*

Andrews et al. [53] (p.1) state that narrative data can seem overwhelming and open to endless interpretation and, by turns, inconsequential and deeply meaningful. Interpretive analysis is a good way to start looking for patterns and meaning. Data is contemporaneously collected and analysed using thematic analysis. Inductive analysis is used to process the data and by looking for recurring themes to make inferences to more general cases. Campbell-Galman [46] (p.22) explains that bottom–up inductive data sorting uses the data itself to make ideas and theories. The data collection is iteratively linked; 'I' poem analysis is employed (Kara [49]) in a cycle of reading, linking, labelling and coding to discover patterns and themes. Barrett and Bolt [10] (p.2) state that creative arts methodologies often situate research in the personal. Student participant Meditrina demonstrates this in a note and poem.

> *I wrote this today, not sure if it's ok but the way I felt the way I wrote! Wishing you a lovely day.* 😁 😄 😄 *Something for someone and something for a bit of kindness. / Time is precious, always remember, so do you! / There was a time were kindness knock your doors, and light greeted with a gentle touch. / I was ready to learn about light, I was there to be the message the one who will perhaps show you the way. / The journey that will lead you to the parallel reality. To the future...to the unknown!*

She does by expressing that she is uncertain about whether this is what was asked for but elaborates that she is trying out an idea and bringing the personal into the light, saying how she feels—hopeful yet tentative—and the use of emojis to add context and expression and emotion to the note before the poem shows that the poem was one way of expressing herself, but that the emojis helped convey the message non-verbally, pictorially (as English is only one of the five languages she uses). This study begins with the personal—participants own stories are told through poems they have written about themselves about their lives. Barrett and Bolt [10] contend that Arts research is experimental like Art and creativity itself, pushing emergent methodologies that may often contradict what is expected of research. The use of poetry as data is unusual, and the use of poetry to analyse data suggested by Kara [49] is also, as she calls it, experimental, emergent and qualitative interpretation.

## 4. Discussion

*4.1. Finding 1: Critical Thinking is a skill*

The first finding to emerge from thematic data analysis on the Poetry Group is that CT can be regarded as a skill, a practice, a muscle that needs developing, and that using questioning CT can increase reflection. Participant Janus states,

> *I think that sort of pushing someone through teaching them in that Socratic way is what really, really makes the deep critical thinker.*

Here, we can see an example of students beginning to make connections for themselves between the development of speaking and listening skills and CT.

Discussion of Finding 1

CT is not just an academic pursuit but like an instrument, as Sennett says, it is to be practiced and developed. A reflection from my field notes describes the journey some participants have travelled. FN, 16/01/19:

> *Meditrina after being so reticent and sitting out of the circle, is inching slowly into the group week by week, Janus says the group has increased her vocabulary, participants are committed to the group. Suggesting, a trip on a Sunday to the slam poetry café, shows a desire to move outside the institution, wanting to motivate themselves outside class time even Terpsichore who has childcare and cultural barriers to overcome was enthusiastic. Participants have suggested creating a zine (a self-published pamphlet, often illustrated). Participants are also suggesting themes for the coming weeks. They report to be using new found skills in their college work, that it is crossing over from the group into course work.*

*4.2. Finding 2: Critical Thinking Promotes Social Conscience and Citizenship*

What came out strongly among the poetry writers is the theme of a social conscience. The idea of self-restraint, having a moral compass, a kind of internalised citizenship, a set of precepts by which one is deemed decent, kind, caring, considerate, creative, not just individually focused but aware of an individual's contribution to the community as a whole. Student participant, Skerion, states,

> *It doesn't mean giving yourself the strictest rules 'you can't do this' and 'you really can't do that' it just means that there are parameters that you're trying to keep stuff within, and you feel like the more it is, it just gives you a better more focused Vision.*

Discussion of Finding 2

Elder and Paul [30] state that CT helps us make right judgement, spot fake news and avoid egocentric, destructive and pathological thought processes which Elder calls dysfunctional thought. By using CT, seeing other points of view, our own wisdom, from experience, can aid us in thinking around and through problems. Gibb [3] states that students should be encouraged to construct knowledge for themselves by using CT.

As a practitioner engaging in a Poetry Group, I have found this research useful. There were times in the research when I was quite surprised by the turn that our discussions took. For example, participant Skerion really developed the theme of regulating one's own behaviour and citizenship saying,

> *It's like the only way to know if there is a line, is to cross it.*

The whole group spoke of enjoying a safe space for creativity and expression of feelings in a non-judgmental arena for ideas and points of view. The recurring theme of a social conscience, the idea of self-restraint, having a moral compass, a kind of internalised citizenship, a set of precepts to determine who and what is deemed decent, kind, caring, considerate, creative, not just individually focused but aware of one's contribution to the community, is important across the research population in the study. Student participant Terpsichore states,

> *Universal themes, maybe all mesh it all together. I think it's definitely something that speaking about things [in poems] that are not spoken about in a really subtle way.*

Here, she is trying to express that, through the self-expression of poetry, she and the group give themselves permission to explore the big questions about life, death, relationships, politics and religion, but it allows for a soft-treading subtlety. Engagement in poetry writing appeared to encourage the participants in the study to see themselves as whole human beings—not just students, but people with emotions, feelings and CT.

### 4.3. Finding 3: Art Students do Enjoy Writing

The kind of writing I ask students to do is very often based on Art history, or I ask them to write reflective notes in their sketchbooks. Very often, students say they do not know what to write. We have plenty of materials to prompt and aid students in these tasks. The poetry group added an oblique strategy, teaching without the formal classroom, in a more peer-mentoring situation, where students did not feel preached to, but felt they were co-authors in the content and delivery of the sessions. When adult lifelong learners own the session, they are invested in their own and each other's learning. At that point, then they begin to enjoy writing.

Discussion on Finding 3

My assumption that Art students do not like to write has proved to be wrong—what adult lifelong learners enjoy is free writing, creative writing. Participant Matuta states,

> *What surprised me about the poetry class is how much I've really taken to it and actually I found it quite infectious.*

This participant loves writing so much that she set herself an extra challenge of writing a poem a day as a New Year's resolution. Participant Janus states,

> *I love it in a way I never believed that I would. I really really enjoy it. I love writing the poem and I love going to the group.*

She enjoys the writing, but it is the added element of being in a group, hearing each other's poems/life stories and being able to tell her own story/poem which seems to be particularly valued.

### 4.4. Finding 4: Critical Thinking through a Community of Inquiry

Through the Community of Inquiry methodology, participants became more thoughtful in their answers, reflective in their thought process, ready to question and not take things at face value—a willingness to delve deeper indicates an increasingly secure self-identity nurtured through engagement in a Community of Inquiry. Creating a Community of Inquiry, Lipman [29] and Biesta [36] agree, may be helpful in developing conditions to support the development of CT. Data from interviews testifies to the ways in which a Community of Inquiry supports the development of emerging relationships; the growth of trust suggests that exposure to CT is embedded in practice and can lead to increased engagement and improvements in achievement for students. There is, however, an element of curation in the design of the group—a level of scaffolding the learners. Participant Matuta is quoted as saying earlier in the article,

> *I think you have to give people the understanding of what critical-thinking actually is*

Participants are FE level-three diploma students; this project aims to develop criticality by giving participants the tools to become independent thinkers. In the Poetry Group, I outlined to student participants what a Community of Inquiry is (theory described in 2.4). I suggested to them that the

Poetry Group would be an ideal Community of Inquiry, as the joint purpose of the group is to increase cultural capital and promote independent thinking through CT. The participants defined themselves as a community by choosing a name for the group, 'The Vernon Street Poets'. As a community, they mentored each other in developing writing and speaking skills and discovered contemporary poetry as well as honing their performance skills.

Discussion of Finding 4

This claim is borne out in evidence from the Poetry Group, which has evolved into a Community of Inquiry with the participants. The Poetry Group appears to have offered a further space for student reflection and a forum in which to talk about the experiences in a safe space where each person can express themselves more fully. Data from the study suggests that this evolution of the Poetry Club to include a Community of Inquiry helped to increase social and cultural capital across the research population and encouraged them to join in and share the big conversations of life. Skerion continues,

> *So I think giving people space to apply critical thinking to what they're doing, just because people feel comfortable with themselves they still got to give themselves rules just about how to behave.*

This suggests that the establishment of the 'ground rules' in a Community of Inquiry coupled with its encouragement of open mindedness and respect for others contributed to the development of social and cultural capital across the research population.

## 5. Conclusion and Recommendations

This article began by asking whether there is a thinking skills deficit as Brown claims and/or whether some approaches to the development of CT can open up new ways of helping students think critically, creatively and for themselves. It also asks what role a Poetry Group could have in developing CT skills for adult lifelong learners in a Further Education setting. It is hoped this article helps to illustrate how the pedagogical interventions employed in this study can support the development of CT, deepen student engagement in learning and improve achievement. The Arts Council [14] (p.3) states that CT promotes imaginative risk-taking, providing solutions to questions and issues within our material, social and virtual worlds. Burke [26] criticises neoliberal ideas that FE is an economic benefit rather than evoking the principles of social justice. Kearney and Diamond [54] state that those who provide FE courses have wider educational, curricula, social and political goals that are misunderstood by other agencies. Well-being and social justice were at the heart of FE in the past, and perhaps they could be again. As educators and policy makers, we could expand our ideas of where education can take place, especially for disadvantaged adult lifelong learners. Learning in informal learning spaces such as community centres, art galleries or coffee shops takes education to where lifelong learners are, creating spaces and situations in which they can learn. This can occur if agencies collaborate to inject lifelong learning and widen participation policy into the Arts curriculum in order to promote criticality and scholarship through mutual engagement in pedagogical interventions such as those employed in this study.

As an educator in the Arts, I need to value my students and their stories. I need to be able to see them as whole people and creators of new knowledge, not passive recipients of other people's knowledge. That is one of the reasons why it is so important for students to be able to think critically. The findings of this study lend support to the claim that opening up pedagogical spaces where teachers of the Arts (or indeed teachers of any other subject or discipline) can treat their students as whole people, encouraging them to treat each other in the same way, can help to create social and cultural capital across groups and support the development of CT. Themes emerging from data generated in the Poetry Group intervention, developed through a Community of Inquiry, lend support to the claim that CT, combined with shared stories of human experience, can connect students in Arts Education to their subject and to each other. In addition, data sets from the study

support the claim that the Poetry Group employed in this research can also be helpful in developing critical and creative thinking among and between students and help in improving educational achievement in Arts Education contexts.

**Author Contributions:** Conceptualization, F. N.; supervision, M.G. All authors have read and agreed to the published version of the manuscript.

**Funding:** This research received no external funding.

**Acknowledgments:** With thanks to the generous giving of time, energy and encouragement from my colleagues and participants.

**Conflicts of Interest:** The authors declare no conflict of interest. The funders had no role in the design of the study; in the collection, analysis, or interpretation of data; in the writing of the manuscript, or in the decision to publish the results.

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
