# Peer review of "The Thinking Skills Deficit: What Role Does a Poetry Group Have in Developing Critical Thinking Skills for Adult Lifelong Learners in a Further Education Art College?"

_education, doi:10.3390/educsci10030073_

Round 1
Reviewer 1 Report
This is an engaging and comprehensive piece of research exploring the potential impact of poetry writing workshops on critical thinking in art students. It has relevance and implications for art pedagogy as well as interdisciplinary learning / teaching more generally.
It is clear the author has considered the ethics, Practicalities and implications of the research. The consideration of the community of inquiry could perhaps include a little more detail. This is indeed a fascinating aspect of the research but it wasn’t clear what was undertaken to generate this community and how it was defined or defined itself. Questions of agency and transgression (hooks) would be interesting here. Section 1.10 would be an opportunity to expand this.
The section about the actual methodology of the workshops could be strengthened. It is never made entirely clear exactly how the workshops were run (there are many different approaches / methodologies here and there’s perhaps not enough depth about which approach wasn’t taken). Some references to reading in approaches to poetry and critical thinking would strengthen this perhaps. Joan Rettallack and Juliana Spahr’s Poetry and Pedagogy is useful. . Section 2.7 does begin to address this but it would be useful to have a great deal specificity about how this worked. How did students interact with the texts chosen? Did students interact at a textual level through erasures / deformations to probe and understand the language? How did students alight on the particular formal aspect they emulated in their own work? Did they choose this themselves or wasn’t it given to them? What happened once they produced the poems? Did they have to make decisions about how they presented them / performed them etc? Did they ever undertake collaborative activity? How did they develop an understanding of the individual’s Understanding of group responsibility? All of these questions radically alter the kind of independent critical thinking that the writing group might engender.
On the whole this is ambitious research that could make a valid contribution to thinking around poetry, critical thinking and interdisciplinary pedagogy.
Author Response
Reviewer 1
Point 1:
a. The consideration of the community of inquiry could perhaps include a little more detail. This is indeed a fascinating aspect of the research but it wasn’t clear what was undertaken to generate this community and how it was defined or defined itself.
b. Questions of agency and transgression (hooks) would be interesting here. Section 1.10 would be an opportunity to expand this.
Response 1:
a. Community of Inquiry theory is outlined and defined in 2.4; 4.4 discusses the findings/ discussion of the Community of Inquiry in the poetry group, here I have demonstrated how the group defined themselves by choosing a group name and peer mentoring each other.
b. in 1.10 I develop hooks’ theme of agency and transgression by describing the use of informal learning spaces, outside teaching time.
Point 2:
a. The section about the actual methodology of the workshops could be strengthened. It is never made entirely clear exactly how the workshops were run (there are many different approaches / methodologies here and there’s perhaps not enough depth about which approach wasn’t taken).
b. Some references to reading in approaches to poetry and critical thinking would strengthen this perhaps. Joan Rettallack and Juliana Spahr’s Poetry and Pedagogy is useful. . Section 2.7 does begin to address this but it would be useful to have a great deal specificity about how this worked.
c. How did students interact with the texts chosen?
d. Did students interact at a textual level through erasures / deformations to probe and understand the language?
e. How did students alight on the particular formal aspect they emulated in their own work?
f. Did they choose this themselves or wasn’t it given to them?
g. What happened once they produced the poems?
h. Did they have to make decisions about how they presented them / performed them etc?
i. Did they ever undertake collaborative activity?
j. How did they develop an understanding of the individual’s Understanding of group responsibility?
k. All of these questions radically alter the kind of independent critical thinking that the writing group might engender.
Response 2:
a. 2.5 outlines how the Poetry Group is run from line 283 – 307. Alternatives that were considered but not used, 247-282.
b. Line 312-319 examined the ideas of Joan Retallack and Juliana Spahr in relation to the poetry group. Section 2.7 has been absorbed into section 2.5 and addresses this point on specificity.
c. The students interact with the text by listening and discussing the poem, , line 310, and responding with their own written poem for homework. Line 307.
d. Line 289, a handout of the poem and a guide to poetic form is given out each week. Some informal discussion around the poem occurred.
e. the emphasis is on informality, poetic form and use of language and theme was discussed, it was left to the student how closely they adhered to style and form. Line 299
f. line 295 poetic for and choice of published poem was decided by myself, them was chosen by the students – line 285
g. line 313, poems are discussed by the group. Possibility of a self-published pamphlet in the future.320.
h. participants at liberty to perform or not, it was optional. 315
i. they have begun to collate poems to create a zine. 321
j. in 4.2 there is a discussion of a finding that participants developed a sense of individual and group citizenship.
k. Line 467, there is some curation of design and scaffolding of learning.
Reviewer 2 Report
It is an interesting project and the author/-s provide a good presentation of what has already been said and they give the results of a small scale project by taking under consideration the conditions in which thinking skills can develop, cultivate and thrive. I suggest the authors to elaborate about the research design and be more explanatory as it concerns the term "observation" (line, 328)
The author/-s use the abbreviation FE (line 13) in abstract without explaining what it means and in the whole paper, either they use Further Education or sometimes they use the abbreviation FE. It is better to use one of them (either the acronym or the whole word)
In line 175, it needs to be a space between the first two words.
Also, see the line 270 (probably unnecessary dot?) and the sentence in line 419 (probably, spelling error).
Author Response
Response to reviewer 2
Point 1:
It is an interesting project and the author/-s provide a good presentation of what has already been said and they give the results of a small scale project by taking under consideration the conditions in which thinking skills can develop, cultivate and thrive. I suggest the authors to elaborate about the research design and be more explanatory as it concerns the term "observation" (line, 328)
Response 1:
Much work has been done in adding to the manuscript in the research design section. Line 277 to 321 outline a much more detailed account of research design. The term ‘observation’ has been removed from the manuscript.
Point 2:
The author/-s use the abbreviation FE (line 13) in abstract without explaining what it means and in the whole paper, either they use Further Education or sometimes they use the abbreviation FE. It is better to use one of them (either the acronym or the whole word)
In line 175, it needs to be a space between the first two words.
Also, see the line 270 (probably unnecessary dot?) and the sentence in line 419 (probably, spelling error).
Response 2:
The abbreviation FE is explained in line 13
Line 175 is corrected adding a space
Line 270 is corrected, the dot is removed.
Line 419 the word ‘some’ has been correctly spelled.
Reviewer 3 Report
The topic of this article is significant. Encouraging the critical thinking is an extremely important part of education in general. The article is well methodologically elaborated and relevant literature was used. It may be necessary to include more recent literature (from 2010 onwards) It is common to use literature up to 5 years old. I recommend that more recent literature be included in the article.
Author Response
Response to reviewer 3
Point 1:
It may be necessary to include more recent literature (from 2010 onwards) It is common to use literature up to 5 years old. I recommend that more recent literature be included in the article.
Response 1:
Removed :
Desjarlais, Robert 1997 Shelter Blues: Sanity and Selfhood among the Homeless. Philadelphia: University of Pennsylvania Press.
Rosaldo, R. Culture & Truth: The Remaking of Social Analysis, Beacon Press: Boston, U.S.A., 1993, pp. 29-30.
Added :
Bell, J. and Waters, S. (2018) Doing Your Research Project. A guide for first time researchers in education, health and social science, 7th edn. Barkshire. Open university Press, Mcgraw Hill Education.
Denscombe, M (2017) The Good Research Guide for small scale social research projects, sixth edn. Berkshire. Open University Press, McGraw Hill Education
Dewey, J. (2018) How We Think, 2nd edn. CreateSpace Independent Publishing platform.
This manuscript is a resubmission of an earlier submission. The following is a list of the peer review reports and author responses from that submission.